# Entropy and Information Theory: Uses and Misuses

**DOI:** 10.3390/e21121170

**Published:** 2019-11-29

**Authors:** Arieh Ben-Naim

**Affiliations:** Department of Physical Chemistry, The Hebrew University of Jerusalem, Edmond J. Safra Campus Givat Ram, Jerusalem 91904, Israel; ariehbennaim@gmail.com

**Keywords:** entropy, second law, thermodynamics, Shannon measure of information, information theory

## Abstract

This article is about the profound misuses, misunderstanding, misinterpretations and misapplications of entropy, the Second Law of Thermodynamics and Information Theory. It is the story of the “Greatest Blunder Ever in the History of Science”. It is not about a single blunder admitted by a single person (e.g., Albert Einstein allegedly said in connection with the cosmological constant, that this was his greatest blunder), but rather a blunder of gargantuan proportions whose claws have permeated all branches of science; from thermodynamics, cosmology, biology, psychology, sociology and much more.

## 1. Introduction

Einstein on Thermodynamics:
“It is the only physical theory of universal content, which I am convinced, that within the framework of applicability of its basic concepts will never be overthrown.”

Einstein on Infinities:
“Two things are infinite: the universe and human stupidity; and I’m not sure about the universe.”

Louis Essen cautions:
“I was warned that if I persisted [criticizing relativity], I was likely to spoil my career prospects.”

Adding that:
“Students are told that the theory must be accepted although they cannot expect to understand it. They are encouraged right at the beginning of their careers to forsake science in favor of dogma.”

This article tells the incredible story of a simple, well-defined quantity having well-defined limits of applicability, which had evolved to reach monstrous proportions, embracing and controlling everything that happens, and explaining everything that is unexplainable.

My main purpose is to deal with the following perplexing question and quandary: How, a simple, well-defined, “innocent” and powerless concept, called inadequately “entropy” had evolved into an almighty, omnipresent, omnipotent, multi-meaning and monstrous entity, rendering it almost God-like which drives everything and is the cause of everything that happens.

Did I say God-like? Sorry, that was an understatement. Entropy has become far more powerful than any God I can imagine!

God, as we all learned in school, is indeed the almighty and the creator of everything, including us. Yet, we also learn that God gave us the freedom to think, to feel, and even the freedom of believing in Him.

Entropy, in contrast has been ascribed powers far beyond those ascribed to God. It not only drives the entire universe, but also our thoughts, feelings and even creativity.

If you think I am exaggerating in saying that entropy has become mightier than God, please read the following quotation from Michael Shermer’s book “Heavens on Earth” [1]:
This leads us to a deeper question: Why do we have to die at all? Why couldn’t God or Nature endow us with immortality? The answer has to do with… the Second Law of Thermodynamics, or the fact that there’s an arrow of time in our universe that leads to entropy and the running down of everything.

These statements are typical nonsenses written in many popular science books. Entropy is viewed not only as the *cause* of a specific process, but a cause for anything that happens including our thinking, feelings and creation of the arts [2,3]. I believe that such a claim is the most perverted view of entropy and the Second Law.

### 1.1. The Unique Nature of Entropy

Entropy is a unique quantity, not only in thermodynamics but perhaps in the entire science. It is unique in the following sense:

There is no other concept in science which has been given so many interpretations—all, except one being totally unjustified. Surprisingly, the number of such misinterpretations still balloons to this day.
(1)There is no other concept which was gravely misused, misapplied and in some instances, even abused.(2)There is no other concept to which so many powers were ascribed, none of which was ever justified.(3)There is no other concept which features explicitly in so many book’s covers and titles. And ironically some of these book though mentioning entropy, are, in fact totally irrelevant to entropy.(4)There is no other concept which was once misinterpreted, then the same interpretation was blindly and uncritically propagated in the literature by so many scientists.(5)There is no other concept on which people wrote whole books full of “whatever came to their mind,” knowingly or unknowingly that they shall be immune from being proven wrong.(6)There is no other concept in physics which was *Equated to time* (not only by words, but by using equality sign “=”), in spite of the fact that it is totally timeless quantity [4,5].(7)There is no other concept in which the role of “cause” and “effect” have been interchanged.(8)Finally, I would like to add my own “no-other-concept” that has contributed more to the confusion, distortion and misleading the human minds than entropy and the Second Law.

If you doubt the veracity of my statement above, consider the following quotation from Atkins’ book [2]:
“…no other scientific law has contributed more to the liberation of the human spirit than the Second Law of thermodynamics.”

The truth is much different:
No other scientific law has liberated the spirit of some scientists to say whatever comes to their minds on the Second Law of thermodynamics!

We shall present *many* similar meaningless and misleading statements about entropy and the Second Law throughout this article.

### 1.2. Outline of the Article

The article is organized in four Sections, as follows: In Section 2 we describe, very briefly, three different definitions of entropy which are equivalent. These definitions are different in the sense that they are derived from completely different origins, and there is no general proof, that they are identical (i.e., that one follows from the other). Nevertheless, they are equivalent in the sense that for whatever process for which we can calculate the entropy changes we obtain agreement between the results obtained by the different definitions.

In Section 3 we present a few formulations of the Second Law. Some of these are “classical” and one is relatively new. We shall present one thermodynamic and one probability formulation of the Second Law.

In Section 4 we discuss some of the most common interpretations of entropy. We shall see that none of these, except one is a valid and provable interpretation.

In Section 5 we shall discuss three directions along which entropy and the Second Law were misused and misapplied. In this Section we discuss the heart of the greatest blunder in the history of science.

The first is the association of entropy with time. The second is the application of entropy to life phenomena, to evolution to social processes and beyond. The third is the application of entropy and the Second Law to the entire universe; not only at present but also at some hypothetical time in the far past (the “birth” of the universe), or in the far future (the so-called thermal-death of the universe).

This paper is essentially a transcription of an invited lecture the author gave in a conference on Thermodynamics and Information Theory, organized by Professor Adam Gadomski, and held in Bydgoszcz, Poland, October 2019. All these topics are discussed in greater details in Ben-Naim [4,5,6,7,8,9].

## 2. Three Different but Equivalent Definitions of Entropy

Here we shall very briefly present three different definitions of Entropy, with more details available in Ben-Naim (henceforth ABN) [4,5,6,7,8,9]. By “different” we mean that the definitions do not follow from each other, specifically, neither Boltzmann’s nor ABN’s definition can be “derived” from Clausius’s definition. By “equivalent” we mean that for all processes for which we can calculate their change in entropy, we obtain the same results by using the three definitions. It is believed that these three definitions are indeed equivalent although no formal proof of this is available.

### 2.1. Clausius’s “Sefinition” of Entropy

Clausius did not really *define* entropy. Instead, he defined a small change in entropy for one very specific process. It should be added that even before a proper definition of entropy was offered, it was realized that entropy is a *state function*. That means that whenever the *macroscopic state* of a system is specified, its entropy is determined. Clausius’ definition, together with the Third Law of Thermodynamics, led to the calculation of “absolute values” of the entropy of many substances.

Let dQ>0 be a *small quantity* of heat flowing *into* a system, being at a given temperature *T*. The *change* in *entropy* is defined as: (1)$$dS=\frac{dQ}{T}$$

The letter *d* here stands for a very *small quantity*, and *T* is the absolute temperature. *Q*, has the units of *energy*, and *T* has the units of *temperature*. Therefore, the entropy change has the units of *energy* divided by units of *temperature*. Sometimes, you might find the subscript “rev” in the Clausius definition, meaning, that Equation (1) is valid only for a “reversible.” This is not necessary; it is sufficient to state that the heat is added in a quasi-static process [4].

The quantity of heat, dQ. *must* be very small, such that when it is transferred into, or out from the system, the temperature *T* does not change. If dQ is a finite quantity of heat, and one transfers it to a system which is initially at a given *T*, the temperature of the system might change, and therefore the change in entropy will depend on both the initial and the final temperature of the system. There are many processes which do not involve heat transfer, yet, from Clausius’ definition, and the postulate that the entropy is a *state function*, one could devise a *path* leading from one state to another, for which the entropy change can be calculated [4].

### 2.2. Boltzmann’s Definition Based on Total Number of Micro-States

Boltzmann defined the entropy in terms of the *total of number accessible micro-states* of a system consisting of a huge number of particles, but characterized by the macroscopic parameters of energy *E*, volume *V* and number of particles *N*.

Consider a gas consisting of *N* simple particles in a volume *V*, each particle’s micro-state may be described by its location vector $R_{i}$ and its velocity vector $v_{i}$, Figure 1. By simple particles we mean particles having no internal degrees of freedom. Atoms such as argon, neon and the like are considered as simple. They all have internal degrees of freedom, but these are assumed to be unchanged in all the processes we discuss here. Assuming that the gas is very dilute so that interactions between the particles can be neglected, then, all the energy of the system is simply the sum of the kinetic energies of all the particles.

Boltzmann postulated the relationship which is now known as the Boltzmann entropy. (2)$$S=k_{B}\log W$$ where $k_{B}$ is a constant, now known as the Boltzmann constant (1.380 × 10^−23^ J/K), and *W* is the number of accessible micro-states of the system. Here, log is the natural logarithm. At first glance, Boltzmann’s entropy seems to be completely different from Clausius’ entropy. Nevertheless, in all cases for which one can calculate changes of entropy one obtains agreements between the values calculated by the two methods. Boltzmann’s entropy, as defined in Equation (2), has raised considerable confusion regarding the question of whether entropy is, or isn’t a subjective quantity [4].

One example of this confusion which features in many popular science books is the following: Entropy is assumed to be related to our knowledge of the state of the system. If we “know” that the system is at some specific state, then the entropy is zero. Thus, it seems that the entropy is dependent on whether one knows or does not know in which state (or states) the system is.

This confusion arose from misunderstanding *W* which is the *total* number of accessible micro-states of the system. If W=1, then the entropy of the system is indeed zero (as it is for many substances at 0 K). However, if there are *W* states and we know in which state the system is, the entropy is still k ln W and not zero!

In general, the Boltzmann entropy does not provide an explicit entropy function. However, for some specific systems one can derive an entropy function, based on Boltzmann’s definition. The most famous case is the entropy of an ideal gas, for which one can derive an explicit entropy function. This function was derived by Sackur [10] and by Tetrode [11] in 1912, by using the Boltzmann definition of entropy. We shall derive this function based on Shannon’s measure of information in the following section.

### 2.3. ABN’s Definition of Entropy Based on Shannon’s Measure of Information

The third definition which I will refer to as the ABN definition. This is Ben-Naim’s definition based on Shannon’s measure of information (SMI). The reader who is not familiar with SMI is referred to [7]. Here we present only a brief account of Shannon motivation, and his definition of SMI, then we outline the definition of entropy based on SMI.

Shannon was interested in a theory of *communication of information*, not information itself. This is very clear to anyone who read through Shannon’s original article [12]. In fact, in the introduction to the book, we find:
“The word information, in this theory, is used in a special sense that must not be confused with its ordinary usage. In particular, information must not be confused with meaning.”

Here is how Shannon introduced the measure of Information:
Suppose we have a set of possible events whose probabilities of occurrence are $p_{1},p_{2},\cdots ,p_{n}$. These probabilities are known but that is all we know concerning which event will occur. Can we find a measure of how much “choice” is involved in the selection of the event or how uncertain we are of the outcome?

If there is such a measure, say, $H\;\left(p_{1},p_{2},\ldots ,p_{n}\right)$, it is reasonable to require of it the following properties:(1)*H should be continuous in the*$p_{i}$*.*(2)*If all the*$p_{i}$*are equal,*$p_{i}=\frac{1}{n_{\prime }}$*then H should be a monotonic increasing function of n. With equally likely events there is more choice, or uncertainty, when there are more possible events.*(3)If a choice be broken down into two successive choices, the original H should be the weighted sum of the individual values of H.

Then Shannon proved the theorem: The only *H* satisfying the three assumptions above has the form:(3)$$H=-K\sum p_{i}\log p_{i}$$

In the following we shall briefly outline the derivation of the entropy of an ideal gas from the SMI. Details may be found in Ben-Naim, Reference [7]. *K*, in the Shannon article is any constant. In application to thermodynamics *K* turns into Boltzmann Constant.

#### 2.3.1. First Step: The Locational SMI of a Particle in a 1D Box of Length *L*

Suppose we have a particle confined to a one-dimensional (1D) “box” of length *L*. We can define, as Shannon did, the following quantity by analogy with the discrete case:(4)H[f(x)]=−∫f(x)logf(x)dx

It is easy to calculate the density distribution, *f*(*x*) which maximizes the locational SMI, H[f(x)] in Equation (4). The result is:(5)$$f_{eq}\left(x\right)=\frac{1}{L}$$

This is a uniform distribution, Figure 2. (6)H(locations in 1D)=logL

#### 2.3.2. Second Step: The Velocity SMI of a Particle in a 1D “Box” of Length *L*

The mathematical problem is to calculate the probability distribution that maximizes the continuous SMI, subject to two conditions, which are essentially a normalization condition and a constant variation.

The result is the normal distribution, see Figure 3: (7)$$f_{eq}\left(x\right)=\frac{\exp\left[-x^{2}/2\sigma^{2}\right]}{\sqrt{2\pi \sigma^{2}}}$$

The subscript eq which stands for *equilibrium* is clarified once we realize that this is the equilibrium distribution of velocities. Applying this result to a classical particle having average kinetic energy $\frac{m<v_{x}^{2}>}{2}$, and using the relationship between the standard deviation $\sigma^{2}$ and the temperature of the system:(8)$$\sigma^{2}=\frac{k_{B}T}{m}$$

We get the equilibrium velocity distribution of one particle in a 1D system:(9)$$f_{eq}\left(v_{x}\right)=\sqrt{\frac{m}{2\pi k_{B}T}}\;\exp\left[\frac{-mv_{x}^{2}}{2k_{B}T}\right]$$ where $k_{B}$ is the Boltzmann constant, *m* is the mass of the particle, and *T* the absolute temperature. The value of the continuous SMI for this probability density is:(10)$$H_{max}\left(velocity\;in\;1D\right)=\frac{1}{2}\log\left(2\pi ek_{B}T/m\right)$$

Similarly, we can write the momentum distribution in 1D, by transforming from $v_{x}\to p_{x}=mv_{x}$, to get:(11)$$f_{eq}\left(p_{x}\right)=\frac{1}{\sqrt{2\pi mk_{B}T}}\;\exp\left[\frac{-p_{x}^{2}}{2mk_{B}T}\right]$$ and the corresponding maximum SMI:(12)$$H_{max}\left(momentum\;in\;1\;D\right)=\frac{1}{2}\log\left(2\pi emk_{B}T\right)$$

#### 2.3.3. Third Step: Combining the SMI for the Location and Momentum of One Particle; Introducing the Uncertainty Principle

We now combine the two results. Assuming that the location and the momentum (or velocity) of the particles are independent events we write:(13)$$H_{max}\left(location\;and\;momentum\right)=H_{max}\left(location\right)+H_{max}\left(momentum\right)=\log\left[\frac{L\sqrt{2\pi emk_{B}T}}{h_{x}h_{p}}\right]$$

Here $h_{x}$ and $h_{p}$ are chosen to eliminate the divergence of the SMI. In writing (14) we assume that the location and the momentum of the particle are independent. However, quantum mechanics impose restrictions on the accuracy in determining both the location *x* and the corresponding momentum $p_{x}$. We must acknowledge that nature imposes on us a limit on the accuracy with which we can determine *simultaneously* the location and the corresponding momentum. Thus, in Equation (14), $h_{x}$ and $h_{p}$ cannot both be arbitrarily small, but their product must be of the order of Planck constant $h=6.626\times 10^{-34}\;J\;s$. Thus, we set:

$h_{x}h_{p}\approx h,$ and instead of (14), we write:(14)$$H_{max}\left(location\;and\;momentum\right)=\log\left[\frac{L\sqrt{2\pi emk_{B}T}}{h}\right]$$

#### 2.3.4. The SMI of a Particle in a Box of Volume *V*

We consider again one simple particle in a cubic box of volume *V*. We assume that the location of the particle along the three axes *x*, *y* and *z* are independent. Therefore, we can write the SMI of the location of the particle in a cube of edges *L*, and volume *V* as:(15)$$H\left(location\;in\;3D\right)=3H_{max}\left(location\;in\;1D\right)=\;3\log\;L\;=\;\log\;V$$

Similarly, for the momentum of the particle we assume that the momentum (or the velocity) along the three axes *x*, *y* and *z* are independent. Hence, we write:(16)$$H_{max}\left(momentum\;in\;3D\right)=3H_{max}\left(momentum\;in\;1D\right)$$

We combine the SMI of the locations and momenta of one particle in a box of volume *V*, taking into account the uncertainty principle. The result is:(17)$$H_{max}\left(location\;and\;momentum\;in\;3D\right)=\;3\log\left[\frac{L\sqrt{2\pi emk_{B}T}}{h}\right]$$

#### 2.3.5. Step Four: The SMI of Locations and Momenta of *N* Independent and Indistinguishable Particles in a Box of Volume *V*

The next step is to proceed from one particle in a box to *N* independent particles in a box of volume *V*. Given the location (x,y,z), and the momentum $\left(p_{x},p_{y},p_{z}\right)$ of one particle within the box, we say that we know the *micro-state* of the particle. If there are *N* particles in the box, and if their micro-states are independent, we can write the SMI of *N* such particles simply as *N* times the SMI of one particle, i.e., (18)SMI(N independent  particles)=N×SMI(one particle)

This equation would have been correct if the micro-states of all the particles were independent. In reality, there are always correlations between the micro-states of all the particles; one is due to the *indistinguishability* between the particles, the second is due to *intermolecular interactions* between the particles.

For indistinguishable particles there are correlations between the events “one particle in $i_{1}$” “one particle in $i_{2}$” … “one particle in $i_{n}$”, we can define the *mutual information* corresponding to this correlation. We write this as:(19)I(1;2;…;N)=logN!

The SMI for *N* indistinguishable particles will then be:(20)$$H\left(N\;particles\right)=\overset{N}{\underset{i=1}{\sum }}H\left(one\;particle\right)-\log N!.\;$$

For the definition of the total mutual information, see Ben-Naim [7].

We can now write the final result for the SMI of *N* indistinguishable (but non-interacting) particles as:(21)H(N indistinguishable particles)=Nlog V(2πmekBTh2)32−logN!

Using the Stirling approximation for logN! (note again that we use here the natural logarithm) in the form:(22)logN!≈Nlog N−N

We have the final result for the SMI of *N* indistinguishable particles in a box of volume *V* and temperature *T*:(23)H(1,2,…N)=Nlog [VN(2πmkBTh2)32]+52N

This is a remarkable result. By multiplying the SMI of *N* particles in a box of volume *V*, at temperature *T*, by a constant factor ($k_{B}\log_{e}2$), one gets the *thermodynamic entropy* of an ideal gas of simple particles. This equation was derived by Sackur [10] and by Tetrode [11] in 1912, by using the Boltzmann definition of entropy. Here, we have derived the entropy function of an ideal gas from the SMI. See also Ben-Naim [7].

One can convert this expression to the *entropy function*
S(E,V,N), by using the relationship between the total kinetic energy of the system, and the total kinetic energy of all the particles:(24)$$E=N\frac{mv^{2}}{2}=\frac{3}{2}Nk_{B}T$$

Hence, the explicit entropy function of an ideal gas is:(25)S(E,V,N)=NkBln[VN(EN)32]+32kBN[53+ln(4πm3h2)]

We can use this equation as a *definition* of the entropy of an ideal gas of simple particles characterized by constant energy, volume and number of particles.

The next step in the definition of entropy is to add to the entropy of an ideal gas, the *mutual information* due to intermolecular interactions. We shall not need it here. The details are available in Reference [7].

## 3. Various Formulations of the Second Law

We shall distinguish between the “*thermodynamic*” formulations and the *probability* formulation of the Second Law. In my opinion the latter is a more general and a more useful formulation of the Second Law. There is a great amount of confusion regarding these two different formulation. For instance, one of the thermodynamic formulations is: when we remove a constraint, from a constrained equilibrium state of an isolated system, the entropy can only increase, it will never decrease. This “never” is in absolute sense. On the other hand, the probability formulation, as we expect, is of a statistical nature; the system will never return to its initial state. Here, “never” is not absolute but only “in practice.”

In all of the examples of irreversible processes given in the literature, it is claimed that one *never* observes the reversal process spontaneously; the gas *never* condenses into a smaller region in space, two gases *never* un-mix spontaneously after being mixed, and heat *never* flows from a cold to a hot body, Figure 4.

Indeed, we *never* observe any of these processes occurring spontaneously in the reverse direction. For this reason, the processes shown in Figure 4 (as well as many others) are said to be *irreversible*. The idea of absolute *irreversibility* of these processes, was used to identify the direction of these process with the so-called Arrow of Time [4]. This idea is not only not true; it is also erroneously associated with the very definition of entropy. One should be careful with the use of the words “reversible” and “irreversible” in connection with the Second Law. Here, we point out two possible definitions of the term *irreversible*:(1)We *never* observe that the final state of any of the processes in Figure 4 returns back to the initial state (on the left-hand side of Figure 4) spontaneously.(2)We *never* observe the final state of any of the processes in Figure 4 going back to the initial state and *staying* in that state.

In case 1, the word *never* is used in “practice.” The system can go from the final to the initial state. In this case, we can say that the initial state will be *visited.*

In case 2, the word *never* is used in an absolute sense. The system will never go back to its initial state and stay there!

In order to *stay* in one chamber, the partition should be replaced at its original position. Of course, this will *never* (in an absolute sense) occur spontaneously.

Notwithstanding the enormous success and the generality of the Second Law, Clausius made one further generalization of the Second Law:
***The entropy of the universe always increases***

This formulation is an unwarranted *over-generalization*. The reason is that the entropy of the universe cannot be defined!

### 3.1. Entropy-Formulations of the Second Law

In this section we shall discuss only one thermodynamic formulation of the Second Law for isolated systems. Other formulations are discussed in [4,6,13].

An isolated system, characterized by a fixed energy, volume and number of particles. For such a system the Second Law states [13]:
The entropy of an unconstrained isolated system (E,V,N), at equilibrium is larger than the entropy of any possible constrained equilibrium states of the same system.

An equivalent formulation of the Second Law is:
Removing any constraint from a constrained equilibrium state of an isolated system will result in an increase (or unchanged) of the entropy.

The maximum entropy is a maximum with respect to all *constrained equilibrium states*. This is very different from the maximum as a function of time, as it is often stated in the literature.

We quote here also the relationship between the change in entropy and the ratio of the probabilities:(26)$$\frac{\mathrm{Pr}\left(final\right)}{\mathrm{Pr}\left(initial\right)}=\exp\left[{\left[S\left(final\right)-S\left(initial\right)\right]/}^{k_{B}}\right].$$

One should be very careful in interpreting this equality. The entropy change in this equation refers to change from state (a) to state (c), in Figure 5, both are equilibrium states. The probability ratio on the left hand side of the equation is for the states (b) and (c). The probability of state (a) as well as of state (c) is one! The reason is that both state (a) and state (c) are equilibrium states.

#### 3.1.1. Probability Formulations of the Second Law for Isolated Systems

We present here these three equations which relate the difference in some thermodynamic quantity, with the ratio of probabilities:(27)$$\begin{array}{c} \frac{\mathrm{Pr}\left(final\right)}{\mathrm{Pr}\left(initial\right)}=\exp\left[{\left[S\left(final\right)-S\left(initial\right)\right]/}^{k_{B}}\right]\\ \frac{\mathrm{Pr}\left(final\right)}{\mathrm{Pr}\left(initial\right)}=\exp\left[{-\left[A\left(final\right)-A\left(initial\right)\right]/}^{k_{B}T}\right]\\ \frac{\mathrm{Pr}\left(final\right)}{\mathrm{Pr}\left(initial\right)}=\exp\left[{-\left[G\left(final\right)-G\left(initial\right)\right]/}^{k_{B}T}\right] \end{array}$$

The first is valid for an isolated system, the second for a system at constant temperature, and the third for a system at constant temperature and pressure.

It is clear from Equations (27) that the probability formulation of the Second Law, which will state below, is far more general than any of the thermodynamic formulations in terms of either entropy, Helmholtz energy or Gibbs energy.

The origin of the probability formulation of the Second law can be traced back to Boltzmann:
“… the system…when left to itself, it rapidly proceeds to the disordered, most probable state.”

Note that here Boltzmann uses the term disorder to describe what happens “when (the system) is left to itself, it rapidly proceeds to disordered most probable state.

#### 3.1.2. The Probability Formulation of the Second Law

Why do we see the process going in one direction? Clearly, the random motion of the particles cannot determine a unique direction. The fact that we observe such a one-way, or one-directional processes led many to associate the so-called Arrow of Time with the Second Law, more specifically with the “tendency of entropy to increase.”

It we ask: “What is the cause of the one-way processes?” or “What drives the processes in one direction?” The answer is probabilistic; the processes are not absolutely irreversible. All these processes are irreversible only *in practice*, or equivalently with *high probability*.

Note that *before* we removed the constraint, the probability of finding the initial state is *one*. This is an equilibrium state, and all the particles are, by definition, of the initial state. However, after the removal of the constrain, we have for *N* of the order of $10^{23}$: (28)$$\frac{\mathrm{Pr}\left(final\;configuration\right)}{\mathrm{Pr}\left(initial\;configuration\right)}\approx infinity$$

This is essentially the probability formulation of the Second Law for this particular experiment. This law states that starting with an equilibrium initial state, and removing the constraint (the partition), the system will evolve to a new equilibrium configuration which has a probability overwhelmingly larger than the initial configuration.

Let us repeat the probability formulation of the Second Law for this particular example. We start with an initial *constrained equilibrium state*. We remove the constraint, and the system’s configuration will evolve with probability (nearly) one, to a new equilibrium state, and we shall *never* observe reversal to the initial state. “Never” here, means never in our lifetime, or in the lifetime of the universe.

#### 3.1.3. Some Concluding Remarks

At the time when Boltzmann proclaimed the probabilistic approach to the Second Law, it seemed as if this law was somewhat *weaker* than the other laws of physics. All physical laws are absolute and no exceptions are allowed. The Second Law, as formulated by Clausius was also absolute. On the other hand, Boltzmann’s formulation was not absolute–exceptions were allowed. Much later came the realization that the admitted non-absoluteness of Boltzmann’s formulations of the Second Law, was in fact more *absolute* than the absoluteness of the macroscopic formulation of the Second Law, as well as of any other law of physics for that matter. It should be noted that although Boltzmann was right in claiming that the system tends to the most probable state, he erred when he claimed that the system tends to a disordered state. More details on this in Ben-Naim [4].

What about the violations of the Second Law? Here, one should be prepared for a shocking statement.

Most writers who write about the Second Law, and who recognize its statistical nature, would say that since the system can return to its initial state, the violation of the Second Law is possible. They might add that such “violations” could occur for small *N*, but it will *never* occur for large *N*. Thus, one concludes that the Second Law, because of its statistical nature can be violated.

I say, No! The Second Law, in any formulation will *never* be violated, never in the absolute sense, not for small *N*, and not for very large *N!*

This sounds “contradictory” to the very statistical nature of the probability formulation of the Second Law. However, it is not. The reason is that the occurrence of an extremely improbable event is not a violation of the law which states that such events have very low probability.

Thus, the system can return to the initial state, in this sense the Second Law is indeed a statistical Law. However, whenever that (very rare) event occurs it is not a violation of the Second Law.

Note also that the Entropy formulation of the Second Law is not statistical. Entropy will not decrease in a spontaneous process in an isolated system. Thus, we can conclude the Second Law, in either formulation, can never be violated, *never* in the absolute sense!

## 4. Interpretation and Misinterpretations of Entropy.

In this section we present only a few misinterpretations of entropy. More details may be found in [4].

### 4.1. The Order-Disorder Interpretation

The oldest and the most common interpretation of entropy, sometimes also used as a “definition” of entropy, is *disorder*. It is unfortunate that this interpretation has survived until these days in spite of being falsified in several books and articles, Ben-Naim [4].

It is not clear who was the first to equate entropy with disorder. The oldest association of disorder with the Second Law is probably in Boltzmann’s writings. Here are some quotations [14,15]:
“… the initial state of the system…must be distinguished by a special property (ordered or improbable) …”
“…this system takes in the course of time states…which one calls disordered.”
“Since by far most of the states of the system are disordered, one calls the latter the probable states.”
“… the system…when left to itself, it rapidly proceeds to the disordered, most probable state.”

In a delightful book *Circularity*, by Aharoni [16], on page 41 we find:
“…disorder (measured by the parameter called entropy) increases with time”

Interestingly, this statement appears in a book on circularity. Here disorder is “*measured by entropy*”, and entropy is a measure of disorder. A perfect example of circularity. I am bringing this quotation from Aharoni’s excellent book, not as a criticism of the book, but as an example of an excellent writer who simply took a wrong idea from the literature and expanded it: On page 42 we find:
“Nobody know for sure why the world is going from order to disorder, but this is so”

This is a typical statement, where one takes for granted the “truth” that the world goes from order to disorder, then expressed only the puzzlement about not knowing why this “fact” is so. Of course no one is sure about that “fact” as no one is sure why the world is going from being beautiful to uglier. The answer is very simple: I am sure that the world is not going from order to disorder. Thus the very assumption made in this quoted statement is definitely not true!

### 4.2. The Association of Entropy with Spreading/Dispersion/Sharing

The second, most popular descriptor of entropy as “spreading” was probably suggested by Guggenheim [17]. Guggenheim started with the Boltzmann definition of entropy in the form: S(E)=klogΩ(E), where k is a constant and: “Ω(E)
*denotes the number of accessible independent quantum states of energy E for a closed system*.”
“To the question what in one word does entropy really mean, the author would have no hesitation in replying ‘Accessibility’ or ‘Spread.’ When this picture of entropy is adopted, all mystery concerning the increasing property of entropy vanishes.”

Authors who advocate the “spreading” interpretation of entropy would also say that when the energy spread is larger, the entropy change should be larger too. Therefore, the change in the entropy in the expansion of one mole of an ideal gas, from volume *V* to 2*V*, in either an isolated system, or in an isothermal process is $\Delta S=R\ln\frac{2V}{V}=R\ln2$ where *n* is the number of moles, *R* is the gas constant. You can repeat the same process at different temperatures, as long as you have an ideal gas, the change in entropy in this process is the same nRln2, independently of the temperature (as well as on the energy of the gas). Thus, as long as we have an ideal gas the change of entropy in the expansion of one mole of gas from *V* to 2*V* is *independent* of temperature. It is always *R*ln2.

### 4.3. Entropy as Information; Known and Unknown, Visible and Invisible and Ignorance

The earliest explicit association between information and entropy is probably due to Lewis [18]:
“Gain in entropy always means loss of information and nothing more.”

It is clear, therefore that Lewis did not use the term “information” is the sense of SMI. The misinterpretation of the information-theoretical entropy as a subjective information is quite common. Here is a paragraph from Atkins’ preface from the book [2] “*The Second Law*”.
“I have deliberately omitted reference to the relation between information theory and entropy. There is the danger, it seems to me, of giving the impression that entropy requires the existence of some cognizant entity capable of possessing ‘information’ or of being to some degree ‘ignorant.’ It is then only a small step to the presumption that entropy is all in the mind, and hence is an aspect of the observer.”

Atkins’ comment and his rejection of the informational interpretation of entropy on the grounds that this “relation” might lead to the “presumption that entropy is all in the mind” is ironic. Instead, he uses the terms “disorder” and “disorganized,” etc., which are concepts that are far more “in the mind.”

### 4.4. Entropy as a Measure of Probability

To begin with, let me say the following: ***Entropy is not probability!***

Numerous authors would start by writing Boltzmann’s equation for entropy as:(29)S=kln P where *P* is supposed to stand for probability. Of course, *P* cannot be probability; Probability, is a number smaller than 1, hence, the entropy would become a negative number.

Brillouin was perhaps the first who wrote Boltzmann’s equation in this form. On page 119 of Brillouin’s book [19] we find:
Entropy is shown to be related to probability. A closed isolated system may have been created artificially with very improbable structure…

Next comes the most absurd, I would also say, *shameful* statement which should not be made by anyone who has any knowledge of probability. On page 120 we find:
***The probability has a natural tendency to increase, and so does entropy. The exact relation is given by the famous Boltzmann-Planck formula:***
S=k ln P

As anyone who knows elementary probability theory, probability does not have (a natural or unnatural) tendency to increase (or to decrease)! Entropy does not have a tendency to increase! And there is no such a relationship between entropy and probability! This is plainly a shameful statement, nothing more.

### 4.5. Entropy as a Measure of Irreversibility

The Second Law is usually discussed in connection with the apparent irreversibility of natural processes. Although the terms reversible and irreversible have many interpretations it is very common to say the that ΔS>0 in the Clausius formulation is a measure of irreversibility.

In a recent book by Rovelli [20] we find on page 25 where the author introduces the entropy as:
Clausius introduces a quantity that measures this irreversible progress of heat in only one direction and, since he was a cultivated German, he gives it a name taken from ancient Greek–entropy.

It is such a shame that a theoretical physicist would write such a sentence in a book published in 2018! To see the fallacy of such a statement, just have a look at Figure 5.

I tell you that in system (a), there are *N* particles at a given volume *V* and temperature *T.* In (c) the same number of particle are in volume 2 *V* and at the same temperature *T*. I will also tell you that the system (c) has higher entropy that system (a). Can you tell which system is more or less “irreversible?”

In a glossary of Lemon’s book [21], we find:
“Entropy: A measure of the irreversibility of the thermodynamic evolution of an isolated system.”

This is not true! First, because irreversibility means several things. Second, because irreversibility is used to describe a process, whereas entropy is a quantity assigned to a *state* of a system. Third, because entropy is a measurable quantity, but “irreversibility," is not a measurable quantity.

Neither the value of the entropy of a system, nor the change in entropy can be a measure of the irreversibility–certainly not the “irreversibility of the thermodynamic evolution of an isolated system.”

Entropy, in itself is a state function. This means that for any well-defined thermodynamic system, say (T,P,N), the entropy is also determined. Specifically, for an isolated system defined by the constant variables E,V,N, the entropy is fixed. The value of S has nothing to do with the “evolution” of the system, nor with any process that the system is going through. 

In the text of Lemon’s book, we find a different “definition.” On page 9, the author states:
“…the entropy difference between two states of an isolated system quantifies the irreversibility of a process connecting those two states”

Indeed, the entropy *difference* is by definition positive for a spontaneous process in an isolated system. However, entropy difference between to states of isolated system, is a quantity fixed by the two states. It does not depend on how we proceed from one state to another; *Reversibility* or *irreversibility* of a process is not quantifiable.

### 4.6. Entropy as Equilibriumness

Here is another amusing idea about the “meaning” of entropy. This “original,” and as far as I know, a unique “invention” appears in Seife’s book “*Decoding the Universe*” [22]. On page 32 of Seife’s book we find the obscure sentence:
“The equilibrium of the universe increases, despite your best effort.”

Can anyone tell me what does it means that the “equilibrium” of anything (let alone the universe) increases (with or without our efforts)? In the next paragraph the author introduces another meaningless term, which is supposed to “explain” the previous obscure sentence:
“What if you don’t use an engine? What if you can turn a crank by hand? Well, in actuality your muscles are acting as an engine, too. They are exploiting the chemical energy stored in molecules in your bloodstream, breaking them apart, and releasing the energy into the environment in the form of work. This increases the “equilibriumness” of the universe just as severely as a heat engine does.”

As I have noted in my book [4], such statements are a result of a profound misunderstanding of the Second Law. There are many other interpretations of entropy, these are discussed in [4].

## 5. Misuses and Misapplications of Entropy and the Second Law

In this section we shall very briefly review three main misuses of entropy and the Second Law. More details in Ben-Naim [4].

### 5.1. The Association of Entropy with Time

The origin of the association of entropy with “Time’s Arrow” can be traced to Clausius’ famous statement of the Second Law:
“The entropy of the universe always increases.”

The statement “*entropy always increases*”, means implicitly that “*entropy always increases with time*”. However, it is Eddington [23] who is credited for the *explicit* association of “The law that entropy always increases” with “Time’s Arrow”, which expresses this “one-way property of time.” See below.

There are two very well-known quotations from Eddington’s book, “*The Nature of the Physical World*” [23]. One concerns the role of entropy and the Second Law, and the second, introduces the idea of “time’s arrow.” In the first quotation Eddington reiterates the unfounded idea that “entropy always increases.” Although it is not explicitly stated, the second alludes to the connection between the Second Law and the Arrow of Time. This is clear from the association of the “*random element in the state of the world*” with the “*arrow pointing towards the future*”. 

There are many other statements in Eddington’s book which are unfounded and misleading. For instance; the claim that entropy is a *subjective* quantity, the concepts of “*entropy-clock*”, and “*entropy-gradient*”. Reading through the entire book by Eddington, you will not find a single correct statement on the thermodynamic entropy! 

### 5.2. Boltzmann’s H-Theorem and the Seeds of an Enormous Misconception about Entropy

In 1877 Boltzmann proved a truly remarkable theorem, known as the H-theorem. He defined a function H(t), and proved that it decreases with time and reaches a minimum at equilibrium. In the following shall we briefly present the H-theorem, the main criticism, and Boltzmann’s answer. We will point out where Boltzmann went wrong and why the function −H(t) is not entropy, and why the H-theorem does not represent the Second Law.

Boltzmann defined a function H(t) as:(30)H(t)=∫f(v,t)log[f(v,t)]dv

Then Boltzmann made a few assumptions. Details of the assumptions and the proof of the theorem can be found in many textbooks. Basically, Boltzmann proved that:(31)$$\frac{dH\left(t\right)}{dt}\leq 0$$ and at equilibrium, i.e., t→∞:
(32)$$\frac{dH\left(t\right)}{dt}=0$$

Boltzmann believed that the behavior of the function −H(t) is the same as that of the entropy, i.e., the entropy always increases with time, and at equilibrium, it reaches a maximum, thereafter it does not change with time.

This theorem drew great amount of criticism. We shall not discuss these here. It is sufficing to say that Boltzmann correctly answered all the criticism, see [4,5]. Notwithstanding Boltzmann’s correct answers to his critics, Boltzmann and his critics made an enduring mistake in the interpretation of the H-function, a lingering mistake that has hounded us ever since. This is the very identification of the H-Theorem with the behavior of the entropy. It is clear from the very definition of the function H(t), that −H(t) is a SMI. If one identifies the SMI with entropy, then we go back to Boltzmann’s identification of the function −H(t) with entropy.

To obtain the entropy one must first define the −H(t) function based on the distribution of both the locations and momentum, i.e.,:(33)−H(t)=−∫f(R,p,t)logf(R,p,t)dRdp

To obtain the entropy we must take the maximum of −H(t) over all possible distributions f(R,p,t): (34)$$Entropy=\underset{over\;all\;fs}{\max}\left[-H\left(t\right)\right]$$

We believe that once the system attains an equilibrium, the −H(t) attains its maximum value, i.e., we identify the *maximum* over all possible distributions with the maximum of SMI in the limit t→∞, i.e., (35)$$Entropy=\underset{t\to \infty }{\lim}\left[-H\left(t\right)\right]=Max\;SMI\;\left(at\;equilibrium\right)$$

At this limit we obtain the entropy (up to a multiplicative constant), which is clearly not a function of time! Thus, once it is understood that the function −H(t) is an SMI and not entropy, it becomes clear that the criticism of Boltzmann’s H-Theorem is addressed to the evolution of the SMI, and not of the entropy. 

In responding the editor’s comment, I would like to add that I am well aware that there exists an extensive literature on non-equilibrium thermodynamics. My conclusion in this article is based on examination of the *definitions* of entropy and the *formulation* of the Second Law. In a separate article submitted to Entropy, I have shown that in all the literature on non-equilibrium thermodynamics all the authors introduce the assumption of “local equilibrium” without giving any justification. I believe that this fact casts some serious doubts on the applicability of thermodynamics to systems which are *very far* from equilibrium.

### 5.3. Can Entropy Be Defined For, and the Second Law Applied to Living Systems?

This question has been discussed by numerous physicists, in particular by Schrödinger, Monod, Prigogine, Penrose and many others [4]. 

One cannot avoid starting with the most famous book written by Schrödinger [24]: What is life?

Chapter 6 titled “Order, disorder and entropy.” He starts with the common statement of the Second Law in terms of the “order” and “disorder”
“It has been explained in Chapter 1 that the laws of physics, as we know them, are statistical laws. They have a lot to do with the natural tendency of things to go over into disorder.”

His main claim is that “living matter evades the decay to equilibrium.” Then he asks:
“How does the living organism avoid decay? The obvious answer is: By eating, drinking, breathing and (in the case of plants) assimilating. The technical term is metabolism.”

I believe the highlight of the book is reached on page 76:
“What then is that precious something contained in our food which keeps us from death? That is easily answered. Every process, event, happening–call it what you will; in a word, everything that is going on in Nature means an increase of the entropy of the part of the world where it is going on. Thus, a living organism continually increases its entropy–or, as you may say, produces positive entropy–and thus tends to approach the dangerous state of maximum entropy, which is death. It can only keep aloof from it, i.e., alive, by continually drawing from its environment negative entropy–which is something very positive as we shall immediately see. What an organism feeds upon is negative entropy.”

Such statements, in my opinion are nothing but pure nonsense. Entropy, by definition, is a positive quantity. There is no negative entropy!

There are many statements in popular science which relate biology to ordering, and ordering as decrease in entropy. Here is an example:

Atkins [2], in his introduction to the book writes:
“In Chapter 8 we also saw how the Second Law accounts for the emergence of the intricately ordered forms characteristic of life.”

Of course, this is an unfulfilled promise. No one has ever shown that the Second accounts for anything associated with life. Brillouin [19], further developed the idea of “feeding on the negative entropy” and claimed that:
“If living organism needs food, it is only for the negentropy it can get from it, and which is needed to make up for the losses due to mechanical work done, or simple degradation processes in living systems. Energy contained in food does not really matter: Since energy is conserved and never gets lost, but negentropy is the important factor.”

In a recent book by Rovelli [20], the nonsensical idea that “entropy is more important than energy, is elevates to highest peak. You will find there a statement written in all capital letters:
IT IS ENTROPY, NOT ENERGY THAT DRIVES THE WORLD.

### 5.4. Entropy and Evolution

In an article entitled: “*Entropy and Evolution*”, Styer [25] begins with a question, “Does the Second Law of thermodynamics prohibit biological evolution?” Then he continues to show “quantitatively” that there is no conflict between evolution and the Second Law. Here is how he calculates the “entropy required for evolution.” Suppose that, due to evolution, each individual organism is 1000 times “more improbable” than the corresponding individual was 100 years ago. In other words, if $\Omega_{i}$ is the number of microstates consistent with the specification of an organism 100 years ago, and $\Omega_{f}$ is the number of microstates consistent with the specification of today’s “improved and less probable” organism, then $\Omega_{f}=10^{-3}\Omega_{i}$.” From these two numbers he estimates the change in entropy per one evolving organism, then he estimates the change in entropy of the entire biosphere due to evolution. His conclusion:
“The entropy of the earth’s biosphere is indeed decreasing by a tiny amount due to evolution, and the entropy of the cosmic microwave background is increasing by an even greater amount to compensate for that decrease.”

In my opinion this *quantitative* argument is superfluous. In fact, it weakens the *qualitative* arguments I have given above. No one knows how to calculate the “number of states” ($\Omega_{i}$ and $\Omega_{f}$) of any living organism. No one knows *what the states* of a living organism are, let alone *count* them. Therefore, the estimated change in entropy due to evolution is meaningless. 

### 5.5. Application of Entropy and the Second Law to the Entire Universe

We already mentioned Clausius’ *over-generalizing* the Second Law. His well-known and well quoted statement:
“The entropy of the universe always increases.”

I do not know how Clausius arrived at this formulation of the Second Law. Clausius did not, and in fact could not understand the meaning of entropy. One can also safely cay that Clausius’ formulation of the Second Law for the entire universe, is not justified and in fact cannot be justified.

In many books, especially popular science books, one finds a statement of the Second Law as:
Entropy always increases.

This statement is sometimes used either a statement of the Second Law or a short version of Clausius’ statement:
Entropy of the universe always increases.

The first statement is obviously meaningless; entropy, in itself does not have a numerical value. Therefore, it is meaningless to say that it increases or decreases.

The second statement seems more meaningful because it specifies the “system” for which the entropy is said to increase. Unfortunately, this is also meaningless since the entropy of the universe is not definable.

On page 299 of Carroll’s book [26], after talking too much about entropy, the author poses a simple question:
“So what is its entropy?”

“Its” refers to the “universe.” Then the author provides some numbers. First, he says that the early universe was “just a box of hot gas,” and a “box of hot gas is something whose entropy we know how to calculate.” No, we do not know! We know how to calculate the entropy of *ideal* gas, i.e., non-interacting particles, i.e., very dilute gas, not extremely dense gas that was supposed to be in the early universe. Then he provides some numbers:$$S_{early}\approx 10^{83}$$

The author explains that the “≈” sign mean: *“approximately equal to,”* and that he wants to emphasize that this is only a rough estimate, not a rigorous calculation:
This number comes from simply treating the contents of the universe as a conventional gas in thermal equilibrium.”

This is *not* an *approximate* number. On the contrary, this is *exactly* a *meaningless number.* The content of the early universe cannot be treated as “conventional gas in thermal equilibrium.” Then, we are given another number of the entropy:$$S_{today}\approx 10^{121}$$ and, yet another number:$$S_{max}\approx 10^{120}$$

All these numbers are meaningless. After finishing with the numbers, the absurdities are taken into new heights. On page 301, we find:
“The conclusion is perfectly clear: The state of the early universe was not chosen randomly among all possible states. Everyone in the world who has thought about the problem agrees with that.”

Everyone? I certainly do not wish to be included as one who has thought about this problem. I never thought about the meaningless entropy of the early universe.

On page 311, we find another “high entropy” absurd question and answer:
“Why don’t we live in empty space?”

I honestly could not believe that a scientist can pose such a silly and laughable question. Then on page 43 of the book we find:
When it comes to the past, however, we have at our disposal both our knowledge of the current macroscopic state of the universe, plus the fact that the early universe began in a low-entropy state. That one extra bit of information, known simply as the “Past Hypothesis,” gives us enormous leverage when it comes to reconstructing the past from the present.

Thus, Carroll, not only assigned numbers to the entropy of the universe at present, and not only estimated the entropy of the universe at the speculative event referred to as the Big Bang, but he also claimed that this low-entropy “fact” can explain many things such as “why we remember the past but not the Future…”

All these senseless claims could have been averted has Carroll, as well as many others recognize that it is meaningless to talk about the entropy of the universe. It is a fortiori meaningless to talk about the entropy of the universe at some distant point in time.

## 6. Conclusions

Starting with the very definitions of entropy it is clear that entropy is a well-defined *state function*. As such it is well-defined for any system at equilibrium. It is also clear that entropy is not a function of time and it is not related to Time’s Arrow. It is also clear that entropy cannot be used for any living system or to the entire universe. As one of the reviewers suggested, the concept of entropy has expanded from heat machines to realms in which it cannot even be defined. This is precisely why I quoted Einstein on thermodynamics. It is true that thermodynamics will not be *overthrown*, provided it is applied “*within the framework of applicability*”.

## Figures and Tables

**Figure 1 entropy-21-01170-f001:**
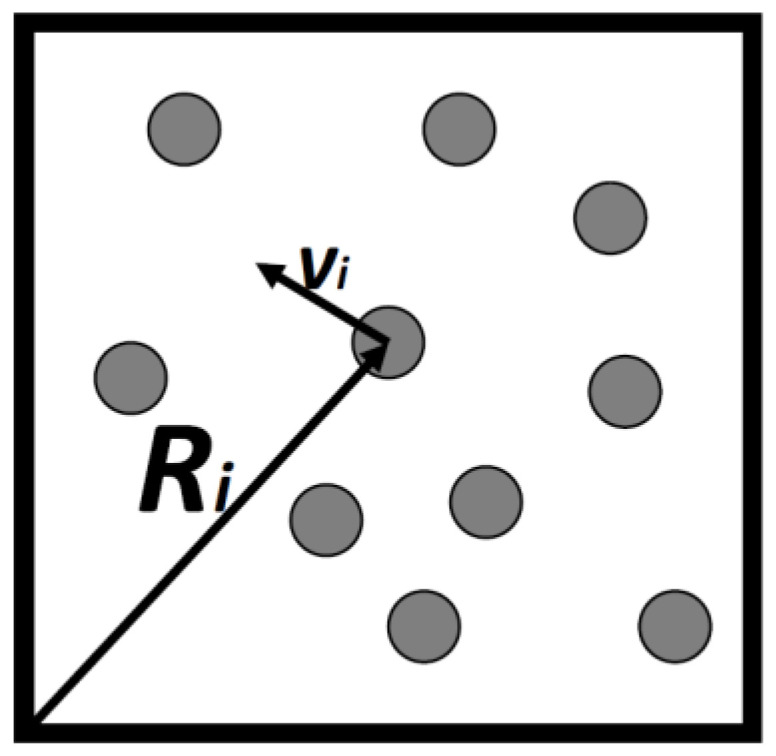
Ten particles in a box of volume *V*. Each particle, *i* has a locational and a velocity vector.

**Figure 2 entropy-21-01170-f002:**
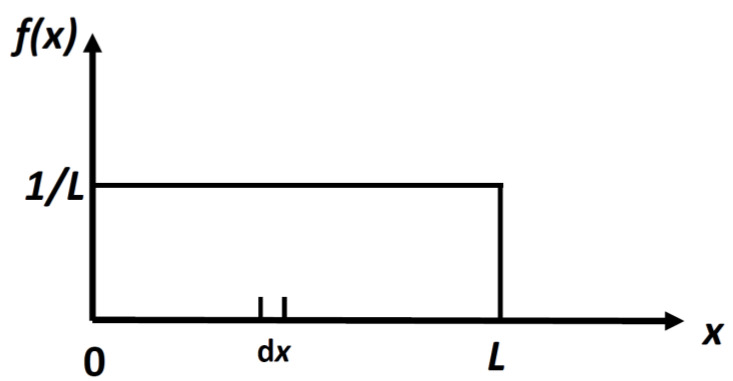
The uniform distribution for a particle in a 1D box of length *L*. The probability of finding a particle in a small interval dx, is dx/L.

**Figure 3 entropy-21-01170-f003:**
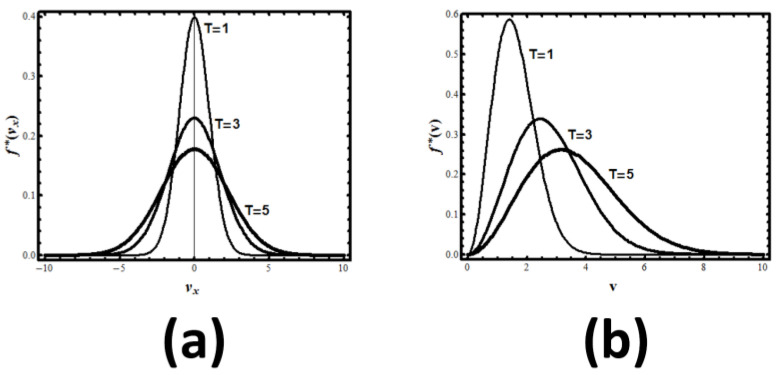
(**a**) The velocity distribution of particles in one dimension at different temperatures; (**b**) The speed (or absolute velocity) distribution of particles in 3D at different temperatures.

**Figure 4 entropy-21-01170-f004:**
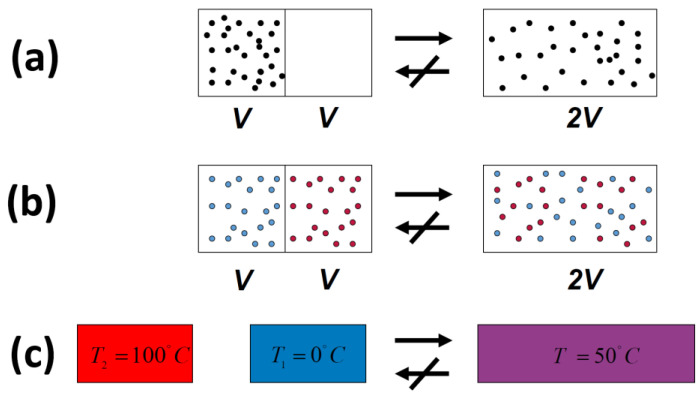
Three typical spontaneous irreversible processes occurring in isolated systems. (**a**) Expansion of an ideal gas; (**b**) Mixing of two ideal gases; (**c**) Heat transfer from a hot to a cold body.

**Figure 5 entropy-21-01170-f005:**
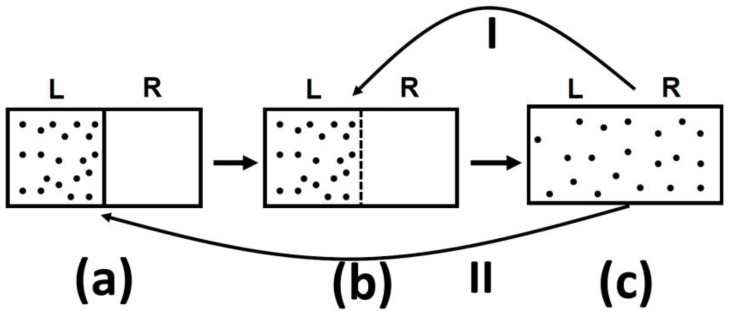
An expansion of an ideal gas. (**a**) The initial state; (**b**) The system at the moment after removal of the partition and (**c**) the final equilibrium state.

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
