# Peer review of "Entropy and Information Theory: Uses and Misuses"

_entropy, 2019, doi:10.3390/e21121170_

Round 1

Reviewer 1 Report

General comments.

I agree with and share the author's view on the misinterpretation and misuse of entropy and the second law of thermodynamics in the popular and scientific literature. The addressed topic is very interesting and the paper has great potential. To be fair, I am not an advocate of Information Theory in the study of physical entropy. I am more comfortable with spreading/dispersion/sharing views of the second law of thermodynamics. But this is not an issue. I think that the paper should be published, but preferably after some extensive revision.

Specific comments.

Form and style: I do not like the style with which the paper has been written. It is too sketchy and it seems a collection of notes written in a hurry. In some parts, the text seems to be too colloquial, but it could be fine if the criticism were more substantiated and not only sketched.

I have a question to the author and the Journal Editor: Is this a scientific article or the author's not-yet-published book promotion? In the first case, I would like a more substantiated criticism (some examples shall follow). In the second case, there is nothing wrong with it, but it is a bit frustrating for the reader to see in the paper a continuous reference to a book not yet published. It is also not so helpful in understanding the author's argumentation. In science-oriented papers, reference is mainly made to already published (possibly peer-reviewed) stuff.

In any case, I would prefer a more stand-alone article. Otherwise, the author should think about the possibility to find a more appropriate place to publish the manuscript. 

-ln 107: Does "ABN", in "ABN's definition", stand for the name of the author? In this latter case, has this definition been previously published in a peer-reviewed journal?

-Section 2.3: As mentioned before, I am personally very suspicious of an information-theoretical approach to physical entropy and the second law. Physics (entropy and the second law are physical concepts) is not information-theory. There could be some analogies in the used mathematical language, but I have reasons to believe that there is nothing more than that. For instance, information-theoretic exorcisms of Maxwell's Demon are highly misleading, a chimera in fact (see, for instance, publications of J. Norton and G. D'Abramo on the topic). But, this is not a problem for the publication of this paper, as long as the author makes the whole paper more readable and with further peer-reviewed references to support his claims.

-lns 290-301: I honestly missed the point here: what does the author want to say?

-lns 374-389: I understand what he wants to say, but it seems to be more "playing with words" than a helpful criticism. I believe that it is sufficiently clear what "violation of the second law" means, even with the statistical formulation. Its is intended not as a violation of the statistical law, but a violation of what the law says to happen 99,9999999999999.....% of the times.

-Section 4.2 & lns 426-433: From a scientific point of view, I like the association of entropy with spreading/dispersion/sharing of energy. And I do not see how the author's criticism at lines 426-433 could be a nail in the coffin for this formulation. The author writes: "Authors who advocate the “spreading” interpretation of entropy would also say (italics mine) that when the energy spread is larger, the entropy change should be larger too". Do they actually say this? Where are the references? Is this a crucial feature for the spreading/dispersion/sharing formulation? Could not entropy be proportional to the "relative" dispersion of energy, namely after normalization with respect to the total energy?

-ln 471: I do not see any Appendix A

-Section 5.2: Boltzmann's H-Theorem is treated vastly in the literature and what the author writes here, with the same quickness adopted for very well known results, is, in fact, something not treated or published before by the same or other authors. Or I am wrong? If I am wrong, it would be nice for the reader to have some references.

Conclusions:

As a matter of fact, there are several (other) issues in the manuscript that, in my humble opinion, should be fixed along these lines. I think that the topic is too interesting to present it in this overall poorly way. I want this article to be published, but I want it to be written in a less sketchy way and I want it to be more stand-alone. The risk is to have readers with frustrated expectations.

Reviewer 2 Report

On introduction:

can be improved:

There is no indication on Carnap entropy concept in the manuscript. The author should provide some insight into this topic. Entropy is basically defined for combustion machines and processes while exhibiting changing states in machines. There were some initiatives to define the entropy of living beings but that would imply the model of living beings as thermodynamic machines which is a clear falsification of life. Neither is the entropy concept acceptable for studying the universe. The universe is not designed as a combustion engine. Therefore, the author should proceed much more carefully with the claims.

On research design:

can be improved:

Please indicate K in formula (2.3) as the total number of communication codes.

Clear presentation of results:

can be improved:

(Part 3) One should clearly make distinction between probability and various arrangements of the system while studying entropy.  On the other hand, one should not omit openness or closeness of the observed system that is studied in the entropy context. Ashby considered some systems as semi-closed.   

(Part 4.3) Information should be quoted as Shannon-Weaver information according to the so-called Information Theory. There is no proper (non tautological) definition neither of information nor of energy because they are fundamental terms – thus they should be taken as they are.

(Part 4.4) Probability is a mathematical construct of the reality, such as a bag with black and red balls. Entropy is an abstraction of the machine behavior. How can anyone compare such items?

(Part 5.2) Entropy is certainly the subject of specific machine constellation, machine arrangement. The specific machine construction should consider physical properties of the engine space, matter, machine kinematics, and dynamics. This can be holistically included into machine functioning as entropy as well, but may not be describable solely with the Boltzmann equation.

(Part 5.3) Entropy is not defined for living beings! Living beings are very far from being considered machines! This is the logic of some weird 19th century biologists. The author should proceed much more carefully with the claims.

(Part 5.4) The universe is definitively not a combustion machine!

Conclusions: must be improved!

Conclusions are not explicitly given. Author presents a forceful criticism on misuse of entropy, which is correct and scientifically brave, especially in biology, and it is acceptable. Science is all about fair and clear cut criticism! The author’s opinion is that Boltzmann concept of entropy is worse than the probabilistic concept.  How can this be justified when applied to machines? To cosmology? Should one construct the answer by studying parallel universes? Author should state that the term entropy has expanded from the machine concept to the cosmology without scientific correctness and put the limits on its correct use. The conclusion should be given with clear indication of major entropy issues to be cleared in biology, cosmology and humanistics. The probability concept may not be the solution to the machine entropy!

Round 2

Reviewer 1 Report

I do not see significative improvements in the text. I am willing to suggest acceptance for publication of the paper as is, provided that the author explicitly declares somewhere in the text that it is a sort of transcription of a conference lecture.

Author Response

My change in the end of the  introduction in RED

This paper is essentially a transcription of an invited lecture, the author gave in a conference on Thermodynamics and Information Theory, organized by Professor Adam Gadomski, and held in Bydgoszcz, Poland, October 2019